# Transbronchial Techniques for Lung Cancer Treatment: Where Are We Now?

**DOI:** 10.3390/cancers15041068

**Published:** 2023-02-08

**Authors:** Joyce W. Y. Chan, Ivan C. H. Siu, Aliss T. C. Chang, Molly S. C. Li, Rainbow W. H. Lau, Tony S. K. Mok, Calvin S. H. Ng

**Affiliations:** 1Division of Cardiothoracic Surgery, Department of Surgery, Prince of Wales Hospital, The Chinese University of Hong Kong, Hong Kong SAR, China; 2Department of Clinical Oncology, Prince of Wales Hospital, The Chinese University of Hong Kong, Hong Kong SAR, China

**Keywords:** lung cancer, transbronchial ablation techniques, robotic bronchoscopy, local treatment, lung cancer screening

## Abstract

**Simple Summary:**

Traditionally, local treatment of lung cancers mainly consists of three branches: surgical resection, radiotherapy, and percutaneous ablation. With the advent of new technologies such as electromagnetic navigation bronchoscopy and robotic bronchoscopy, transbronchial therapies are being developed. This gives substantial hope to high-risk patients, especially those who have had prior chest radiation exposure or frail patients who could not tolerate surgery, as they would otherwise be excluded from the currently established forms of local treatment. In addition, demand for local ablation is also rising due to an increasing incidence of multiple synchronous lung cancers arising in the same patient. In this review, we discuss state-of-the-art transbronchial techniques for lung cancer treatment.

**Abstract:**

The demand for parenchyma-sparing local therapies for lung cancer is rising owing to an increasing incidence of multifocal lung cancers and patients who are unfit for surgery. With the latest evidence of the efficacy of lung cancer screening, more premalignant or early-stage lung cancers are being discovered and the paradigm has shifted from treatment to prevention. Transbronchial therapy is an important armamentarium in the local treatment of lung cancers, with microwave ablation being the most promising based on early to midterm results. Adjuncts to improve transbronchial ablation efficiency and accuracy include mobile C-arm platforms, software to correct for the CT-to-body divergence, metal-containing nanoparticles, and robotic bronchoscopy. Other forms of energy including steam vapor therapy and pulse electric field are under intensive investigation.

## 1. Introduction

Lobectomy has remained the gold standard treatment for lung cancer for over 20 years, as evidenced by the landmark Lung Cancer Study Group trial in 1995 which reported higher mortality and recurrence rates associated with sublobar resection compared to lobectomy [1]. However, increasing numbers of small lung nodules are being incidentally discovered on computer tomography (CT) scans. Many of these (up to 50%) harbor premalignant or very early-stage lung cancers without regional spread, potentially eradicable with other less invasive forms of local treatment rather than surgery. This phenomenon is fueled both by the increasing availability of CT scans worldwide and the established evidence of lung cancer screening in high-risk populations. The National Lung Screening trial reported 15–20% reduction in lung cancer-specific mortality in 55–74-years-old smokers when low-dose helical CT scans were performed in contrast to chest X-rays [2]. More recently, the NELSON trial demonstrated a decrease in mortality of 26% in high-risk men and up to 61% in high-risk women over a 10-year period by comparing low-dose CT screening to no screening [3]. Of note, the trial reported more early-stage lung cancers discovered in the screened group, which may become the best candidates for less invasive forms of treatment. In addition, it is now not uncommon to encounter patients who have had major lung resection for primary lung cancer in the past and are being followed up for growing and increasing solidity of suspicious lung nodules in the remaining lobes [4]. Very often, in South East Asia, biopsy of these lesions would show different genetic clonality from the previous cancer, indicating that these are patients with a predisposition to multifocal lung cancers due to either genetic or environmental causes [5]. This gives rise to the need of lung parenchyma-sparing techniques, for example, sublobar resection, stereotactic body radiation therapy (SBRT), and local ablation [6]. Furthermore, the aging population and improved medical care has also augmented the number of elderly people being diagnosed with lung cancer, many of whom have multiple comorbidities, inadequate lung function, and a high clinical frailty score, making them unsuitable for major pulmonary resection and, hence, better candidates for local treatment.

## 2. Novel Concept in Lung Cancer Prevention and Treatment

There has also been a concept change or paradigm shift from treatment to prevention of lung cancers in recent years. Similarly to the adenoma–carcinoma sequence of colorectal cancers, studies have shown that lung adenocarcinomas follow a similar sequence of progression from premalignant lesions (atypical adenomatous hyperplasia) to adenocarcinoma in situ or minimally invasive adenocarcinoma and eventually to invasive adenocarcinoma [7,8,9]. A significant proportion of lung cancers, especially adenocarcinomas, progress in a stepwise manner while accumulating gene mutations, thus offering an opportunity for early treatment and prevention of invasive disease. Moreover, the degree of invasiveness can be predicted by the consolidation-to-tumor ratio (>0.25) and size of lesions (>2 cm) on CT scans [10], thus offering a basis for the biopsy and treatment of the highest-risk nodules in patients with multiple lung lesions. Similarly to the concept of polypectomy or endoscopic submucosal resection for premalignant or early colorectal cancers, less invasive means of local treatment may also be employed for premalignant or minimally invasive lung cancers. The recently published JCOG 0802 trial is a phase III randomized controlled trial comparing segmentectomy versus lobectomy in peripheral non-small-cell lung carcinomas (NSCLC) less than 2 cm in diameter and >0.5 consolidation-to-tumor ratio. It showed that the five-year overall survival was higher in the segmentectomy group than in the lobectomy group (94.3% vs. 91.1%, *p* = 0.0082), the forced expiratory volume in 1 s was higher in the segmentectomy group, while the local relapse rate was lower in the lobectomy group (5.4% vs. 10.5%) [11]. The CALGB 140,503 investigators have recently further reported noninferiority of sublobar resection (59% wedge resection and 41% segmentectomy) with respect to disease-free survival and overall survival in stage 1 NSCLC < 2 cm [12].

## 3. Transbronchial Microwave Ablation

There is increasing evidence for local treatment of early lung cancers, especially in patients who decline surgery or those with high surgical risks. Evidence shows that sublobar resection confers similar five-year survival rates, especially in older patients, patients with tumor diameter less than 2 cm and/or pure bronchoalveolar carcinoma [13,14,15]. Meanwhile, SBRT is indicated for patients with stage I or II NSCLC without lymph node involvement and who are surgically inoperable. In multiple retrospective series, SBRT has a decent local control rate of more than 80% [16] and a disease-free survival of 26%. In a multicenter phase II study, an overall survival of 40% at four years has been reported [17]. However, both sublobar resection and SBRT still carry risks such as intra- and postoperative surgical complications and radiation pneumonitis, respectively. Percutaneous ablation of lung tumors has been attempted since the early 2000s [18] following the success of local ablation in hepatocellular carcinomas. The subsequent years have seen the popularization of image-guided local ablation therapies of lung tumors, the first one being radiofrequency ablation (RFA), followed by microwave ablation (MWA) and then cryoablation.

In contrast to the percutaneous route, transbronchial ablation has gained popularity in recent years. A bronchoscopy-guided cooled RFA technique targeted towards lung cancers in human subjects was first pioneered by a Japanese group [19,20]. This was followed by a group in China using electromagnetic navigation bronchoscopy (ENB) guidance [21,22]. The advantage of the transbronchial route over the percutaneous route is the avoidance of pleural puncture, hence fewer pleura-based complications. The data from the Japanese group suggested no pneumothorax, bronchopleural fistula, or pleural effusion in twenty-eight cases of transbronchial RFA [20]. In contrast, the rate of pneumothorax due to percutaneous ablation ranges from 3.5% to 54%. Furthermore, other advantages of transbronchial ablation are the avoidance of needle tract seeding and the ability to reach certain parts of the lung which are difficult or even dangerous with percutaneous access. Examples include areas near the diaphragm, lung apex, mediastinal pleura, or areas shielded by the scapula.

Our institute was one of the first to perform microwave ablation with ENB guidance in a hybrid operating room [23]. Microwave energy, unlike radiofrequency, heats tissue directly to lethal temperatures ≥ 150 °C through dielectric hysteresis [24]; therefore, it is independent from electrical conductance. MWA is able to produce faster, larger, and more predictable ablation zones than RFA [25] since microwave energy deposition is less affected by tissue impedance, which is especially high in aerated lungs. The patients treated with this novel technique at our center had elevated surgical risks, due to either insufficient respiratory reserve or underlying comorbidities. Ablated nodules were less than 3 cm in size and away from large blood vessels, which may have led to underablation due to the heat sink effect, and from structures that may suffer from thermal injury, including the phrenic nerve, brachial plexus, heart, and esophagus.

Navigation accuracy has been much improved following the advent of ENB with the support of navigation systems such as SuperDimension^TM^ (Covidien, Plymouth, MN, USA) and further fine-tuned by position confirmation by fluoroscopy and cone beam CT. A microwave catheter (Emprint^TM^ ablation catheter with the Thermosphere^TM^ technology, Covidien, Plymouth, MN, USA) is inserted within the lung tumor transbronchially and ablated for up to 10 min per burn. For larger tumors, double or triple ablation either in the same position or with catheter position adjustment have been performed to ensure adequate margins.

The Navablate study, in which our institute participated, reported the results of 30 nodules undergoing transbronchial MWA. The median nodule size was 12.5 mm, the procedure day technical success rate was 100%, the mean ablative margin was 9.9 mm, and the technique efficiency (satisfactory ablation as evidenced by a one-month CT scan) was 100% [26]. The technique was safe, with only a 3.3% rate of device-related adverse events (mild hemoptysis), and there were no serious adverse events over 30 days [26]. Since early 2019, the author’s institute has accumulated 122 cases with 100% technical success rate [27] (Figure 1). The median length of stay was one day only, similar to that reported for the percutaneous approach. Only 3.4% developed pneumothorax requiring chest drainage, and there were two cases of bronchopleural fistula both treated with an endobronchial valve [28]. Post-ablation reaction and fever occurred in 8.9%, minor hemoptysis or hemorrhage—in 4.4%, pleural effusion—in 1.7%, and chest or pleural space infection—in 2.6% [27]. Since our early results were published in 2021 [23], updated midterm data have shown five cases (4.0%) of local ablation site recurrence over a median follow-up of 507 days. On the other hand, a Chinese group who performed transbronchial MWA in 13 patients with different instruments reported a complete ablation rate of 78.6%, a two-year local control rate of 71.4%, and a median progression-free survival of 33 months [29,30]. The systems used by the Chinese colleagues were a flexible water-cooled MWA antenna (Vison-China Medical Devices R&D Center) connected to a microwave platform (Surblate, Vison) and an MWA device for ablation by Nanjing Nisionmedic (Nanjing, China).

A major limitation of ablation treatment is the lack of preoperative pathological examination in some of the lesions in our series. Ideally, tissue sampling to confirm malignancy or at least premalignant status is preferable prior to ablation. In our institute, CT-guided biopsy or bronchoscopic biopsy for nodules of more than 1 cm in size is attempted whenever possible before ablation. In a handful of cases, we performed same-session ENB transbronchial biopsy followed immediately by ablation if the frozen section showed malignancy. However, this approach presents a particular difficulty in determining the post-ablation margin since multiple biopsies induce perilesional hemorrhage which blurs the border of the original nodule. In addition, on a post-ablation CBCT, it may be difficult to differentiate whether the ground glass changes are caused by hemorrhage or by ablation. An alternative solution is to perform ENB biopsy and ablation in separate sessions, although the patients’ experience may be sacrificed as they have to undergo the procedure twice. Therefore, another approach to this dilemma is to assess the probability of malignancy of target nodules by well-established models, for example, the Herder risk model recommended by the British Thoracic Society, and direct ablation without biopsy is offered to those with a high probability of cancer.

However, the ablation workflow is far from perfect, and further improvements are being developed to smoothen the procedure and reduce the radiation dose. The average number of CBCTs per nodule ablation is 7.1 in our case series [23], yet the minimum number of CBCTs required is only four (pre-procedure, needle position confirmation, ablation catheter position confirmation, post-ablation). The discrepancy is mainly due to incorrect navigation due to the CT-to-body divergence, i.e., the discrepancy between the static preoperative planning CT scan and the dynamic breathing lung, requiring renavigation or redeployment of the needle. The Illumisite^TM^ platform is the first navigational system on market to correct for the CT-to-body divergence. This digital tomosynthesis-based navigation correction is achieved by utilizing fluoroscopic navigation technology which creates a 3D image enhancing the nodule’s visibility so that the adjusted nodule position can be updated. Throughout the procedure, it provides real-time confirmation that the catheter is aligned to the nodule, and this alignment can be maintained even after the locatable guide (LG) is removed, owing to the presence of position sensor coils embedded in the tip of the extended working channel (EWC) (Figure 2). These continuous positional data allow for multidirectional sampling and more thorough biopsy, as demonstrated by biopsy series using this platform [31,32].

The improved navigation and accuracy provided by this platform may also simplify the workflow of transbronchial microwave lung ablation. In previous models without extended working channel (EWC) positional electronics, after the LG is removed, the EWC often returns to its original curvature; thus, subsequent needle deployment is often off-center. With the continuous positional data provided by the tip of an Illumisite^TM^ EWC, a CrossCountry^TM^ needle can be advanced directly at the green ball in the most bullseye view. Moreover, contrary to previous platforms such as the SuperDimension^TM^ navigation system, apart from the direction, the exact distance between the center of the nodule and the tip of an EWC is available, too, allowing precise placement of the tip of the ablation catheter (Figure 2B,C). Since the center of ablation is 1 cm proximal to the tip of an Emprint^TM^ ablation catheter, the EWC is advanced 1 cm beyond the center of the lesion and the ablation catheter is placed flush to the EWC so that the center of ablation coincides with the center of the lesion. Fewer CBCTs are required for confident catheter placement, translating into reduced radiation exposure. The first-in-the-world ENB microwave ablation using the Illumisite^TM^ fluoroscopic navigation platform was successfully performed in mid-2022 [33], and a handful of cases has been subsequently performed at our center without major complications.

To further expand the scope of transbronchial ablation, mobile C-arm machines are available to provide high-quality intraoperative reconstructed CT images at institutes without built-in hybrid theatres. Examples include Cios Spin^®^ by Siemens Healthineers [34,35] (Figure 3) and the O-arm^TM^ O2 imaging system by Medtronic [36]. In a case series comprising 10 cases utilizing the former system, the navigation success was 80%, while the diagnostic yield for biopsy was 80% [37]. Mobile C-arms had a comparable ease-of-use to floor-mounted CBCT machines and were able to identify most lung lesions, except for two sub-centimeter pure ground glass nodules. However, in order to expand its use to lung nodule ablation, further marking tools, image overlay, and segmentation capabilities are required to ablate nodules that are difficult to visualize on fluoroscopy [37]. Recently, Ion^TM^ by Intuitive has integrated Cios Spin^®^ within its own robotic bronchoscopic platform, enabling automatic image transfer from the mobile C-arm machine to an Ion^TM^ bronchoscope during procedures [38]. With the increasing popularity of these systems, it is expected that transbronchial treatment can be expanded to less affluent institutions without hybrid operating rooms so as to become more readily available to patients.

Ablation zone size is the major limiting factor for the types of early lung tumors that can be treated with transbronchial ablation. The typical CT appearance of early post-ablative morphology of concentric ground glass opacities (GGO) actually contains an outer rim of denser GGO containing congested lung tissue which remains viable [39]. The size observed on CBCT tends to overestimate the area of true coagulation necrosis by 4.1 mm [40]. Thus, a minimal margin of at least 5 mm on a CT scan is recommended to assure adequate tumor kill [40,41,42]. For the Emprint^TM^ ablation catheter, the maximum single ablation produces a predicted oval ablation zone of 4.2 cm in length and 3.5 cm in diameter; thus, the maximum size of a lung nodule taking into account a 5 mm margin on both sides is 2.5 cm. The upper limit of nodule size can be extrapolated to 3 cm if double or triple ablation is planned. Upon analyzing our early experience in MWA, the mean minimal ablation margin is 5.51 mm, and a smaller-than-predicted ablation zone size is not significantly associated with the presence of emphysema, solidity of the lesion, or presence of ≥3 mm diameter blood vessels within 5 mm of the lesion. Further studies are required to investigate the factors at play influencing the actual size of the ablation zone. Apart from improving the design of ablation coils to produce a wider ablation zone, another potentially effective way is to infiltrate the tumorous region with metallic nanoparticles. Nanoshells, typically composed of gold as the metallic component and silica as the dielectric material [43], may theoretically be injected into the tumor tissue and augment the effect of microwave heating as metals reflect microwaves.

## 4. Robotic Bronchoscopy to Potentially Improve Transbronchial Ablation Accuracy and Efficacy

Robotic-assisted bronchoscopy (RAB) was originally developed to improve the yield and accuracy of lung nodule biopsy. In comparison to conventional bronchoscopy or other forms of guided bronchoscopy, including virtual bronchoscopy (VB), radial endobronchial ultrasound (r-EBUS), and electromagnetic navigation bronchoscopy (ENB), robotic-assisted bronchoscopy allows a deeper navigation depth up to the ninth airway generation in addition to direct visualization of airways [44]. Currently, there are two major players on the market, the Auris Health Monarch^TM^ platform by Ethicon [45] and the Ion platform by Intuitive [46]. The Monarch^TM^ RAB consists of a telescopic mother–daughter configuration with a 6 mm outer sheath and a 4.2 mm inner scope, utilizing electromagnetic navigation and peripheral vision for guiding navigation (Figure 4). The inner bronchoscope has superior maneuverability to conventional bronchoscopes due to a four-way steering control. While the inner scope is being advanced, structural support is provided by the RAB system, hence reducing the likelihood of scope prolapse into proximal airways. The Ion RAB comprises an articulating catheter with a 3.5 mm outer diameter and a thin 1.8 mm removal visual probe, employing fiberoptic shape sensing for peripheral navigation. The shape-sensing technology reconstructs and displays the entire shape of a thin flexible optical fiber with ultrahigh speed feedback, providing operators with unprecedented three-dimensional perspective on the location of the tip of the catheter. The shape-sensing technology allows positional adjustments compensating for a degree of deformation created when relatively rigid instruments pass through the bronchoscope.

In cadaveric and early human trials of RAB, up to 88–97% navigational success and 77–97% diagnostic yield [47,48,49,50,51,52,53] were reported, particularly when combined with radial EBUS. In lesions without the bronchus sign, the diagnostic yield was higher than that reported using previous nonrobotic systems (54% vs. 31–44%, respectively), such as the SuperDimension^TM^ navigation system [54]. The latest interim results from a multicenter observational real-world robotic-assisted bronchoscopy biopsy study (TARGET study) reported a 97.5% navigational success and a 91% nodule localization by radial EBUS, while the diagnostic yield data have yet to be published [55]. RAB provides the ability to control the distal end of the scope through multiple active articulation points of the scope, thus allowing enhanced instrument maneuverability (Figure 5). The addition of RAB to transbronchial ablation likely streamlines the navigation process, and in combination with cone beam CT in a hybrid operating room, it further improves the accuracy of ablation catheter placement (Figure 4). A prospective multicenter single-arm study using the Neuwave Flex microwave ablation system and the Auris Monarch platform is currently underway (POWER study).

## 5. Other Ablative Energies and Adjuncts

Cryoablation has been performed since the mid-2000s in a percutaneous manner for lung nodules. Temperature can reach up to −40 °C when pressurized argon gas expands [56]. Cryoablation carries multiple advantages: the size of ablation zones is generally larger and independent from impedance, bronchovascular structures are better preserved, no tissue charring is created, and excellent visibility of the iceball on CT is obtainable [57,58,59]. However, the major limitation is the lengthy time per freeze–thaw–freeze cycle, approximately 25 vs. 10 min for cryoablation and MWA, respectively [60]. To date, there is no commercially available transbronchial cryoablation model on the market, although early studies in in vivo porcine models demonstrated satisfactory coagulative necrotic zone and safety [61,62].

Bronchoscopic thermal vapor ablation (BTVA) is an established modality for minimally invasive lung volume reduction in severe emphysema [63]. In 2021, this technique was first performed in humans for lung cancer. In a single-arm treat-and-resect feasibility study [64], 330 Cal thermal vapor energy was delivered to target segments containing lung cancer, and four out of five patients developed large uniform ablation zones, with complete or near-complete necrosis of the target lesions in two patients upon lobectomy five days after vapor ablation. There were no major procedure-related complications, although one case had a significant number of remaining viable tumor cells due to nonuniform ablation zones. Further studies are required to explore the reason for this nonuniformity and improve on it before it can be reliably used for cancer treatment. BTVA may be inadequate as the sole cancer treatment modality, but it can be used as an adjunct to MWA. Conceptually, a cancerous nodule can be ablated transbronchially with microwaves while the margins, satellite nodules, and spread-through airspaces surrounding the nodule can be captured by thermal vapor ablation afterwards, thus potentially further lowering the local recurrence rate.

Pulsed electric field (PEF) is a nonthermal ablative modality that uses a short-living strong electrical field created around a catheter to create microscopic pores in cell membranes (electroporation). When the induced transmembrane voltage crosses the threshold, changes in membrane permeability are observed, presumably related to the formation of hydrophilic pores in the lipid membrane. The degree of electroporation depends on pulse amplitude, pulse duration, number of pulses, and pulse frequency, and the adjustment of these factors has different effects on different tissues. This technology is currently receiving high interest in cardiac ablation for arrythmia [65] as its advantage over the currently available ablation modalities is that tissue destruction is largely nonthermal and tissue-selective (without causing pulmonary vein stenosis). Furthermore, energy can be delivered quickly in seconds or minutes. In murine models, nanosecond pulsed electric field induced hepatocellular carcinoma cell death, increased its phagocytosis by human macrophage cells by eliciting a host immune response against tumor cells [66]. PEF does not thermally denature proteins, allowing the release of intact tumor-associated antigens for antigen-presenting cells to drive a tumor-specific response, thus promoting an improved local response and abscopal effect [67]. Safety of transbronchial delivery of pulsed electric fields in lungs has been reported in swine models. Histologically, short-term effects indicate a thin band of reactive changes at periphery-bounding hemorrhagic regions, with long-term effects presenting ongoing fibrosis and reparative processes. Histology revealed a well-demarcated treatment effect and structurally intact major airways and vessels, without pleural disruption [68]. There is also postulation that lesions treated with PEF may have increased PD1 expression allowing synergistic treatment with anti-PD1 checkpoint blockade [69,70]. The Aliya system (Galvanize Therapeutics, GTI-00018 investigational device, San Carlos, CA, USA) is a biphasic monopolar PEF system which is currently undergoing a clinical trial in human lungs [71].

## 6. Conclusions

There is a rising need for parenchyma-sparing local therapies for lung cancer due to the increasing availability of CT screening, incidence of multifocal lung cancers, and the aging population where a larger proportion of patients are becoming unfit for surgery. Lung cancer screening has driven the discovery of more small-sized premalignant or early-stage lung cancers, and the strategy to tackle these nodules has shifted from treatment to prevention. Transbronchial therapy is an important armamentarium in the local treatment of lung cancers. Compared to other forms of thermal energy, microwave ablation is the most promising one based on the currently available early to midterm results. Numerous adjuncts to improve transbronchial ablation efficiency and accuracy have been developed. Mobile C-arm platforms with high-resolution CBCT allows transbronchial ablation to be performed at institutions without hybrid operating room. Software to correct for the CT-to-body divergence is currently available to enhance biopsy and ablation accuracy and thereby limit radiation exposure. Robotic-assisted bronchoscopy allows fine adjustment of the catheter tip with in-built proprioception to improve the precision of nodule puncture. Other forms of energy including steam vapor therapy and pulse electric field are under intensive investigation. The arena of transbronchial strategies for lung cancer treatment is certainly an area of hot research with promising future.

## Figures and Tables

**Figure 1 cancers-15-01068-f001:**
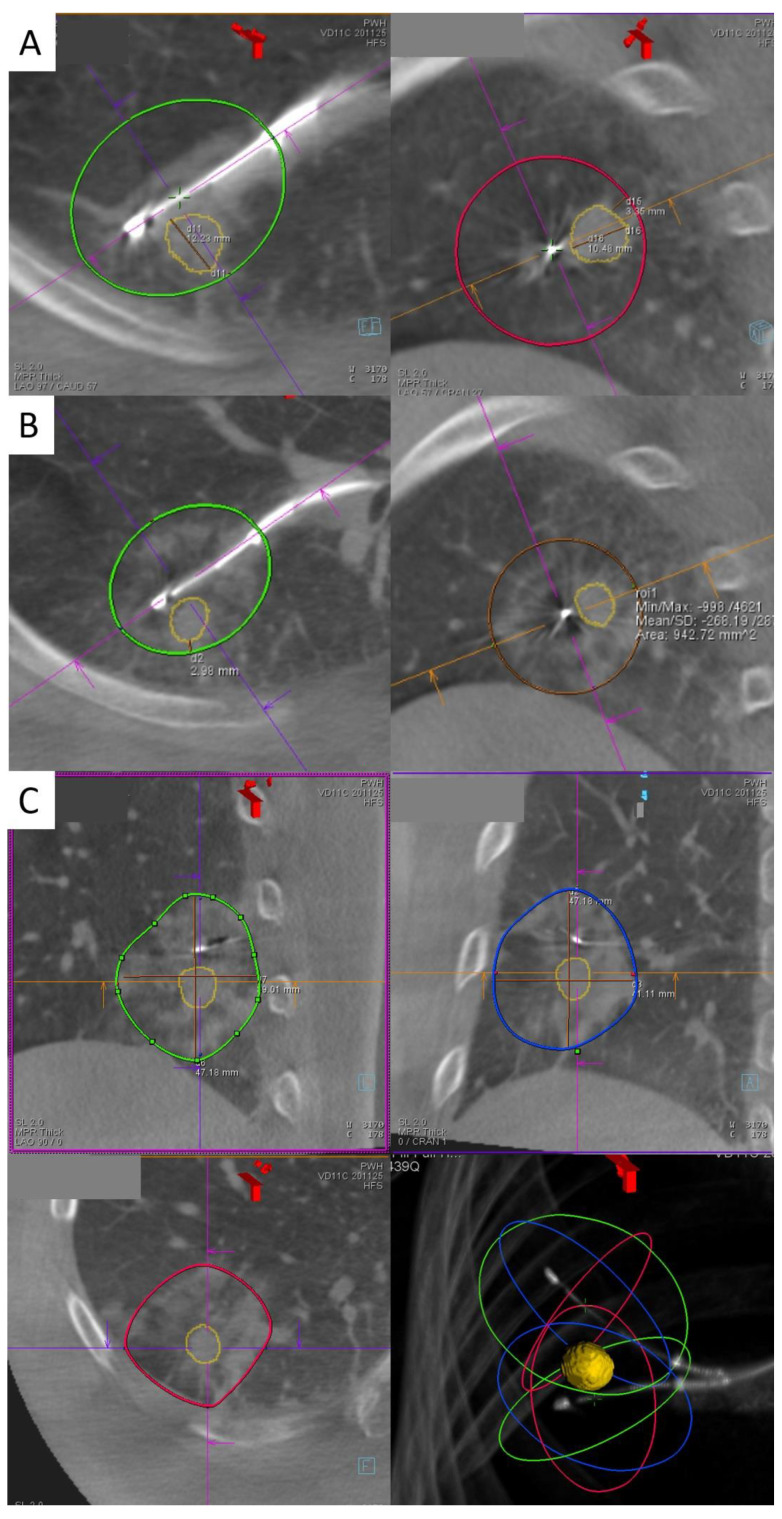
(**A**,**B**) Right lower lobe lung lesion before (**A**) and after (**B**) the first transbronchial microwave ablation in two axes on a cone beam CT scan. The lesion is marked by orange tracings while the green and red ovals represent the predicted ablation zone. In (**B**), ablative changes represented by ground glass opacities are seen surrounding the ablation catheter as predicted; however, the margin was only 2 mm. Therefore, repositioning of the ablation catheter and re-ablation were performed as shown in (**C**), and the final cone beam computer tomography (CBCT) shows a satisfactory margin of 6.7 mm.

**Figure 2 cancers-15-01068-f002:**
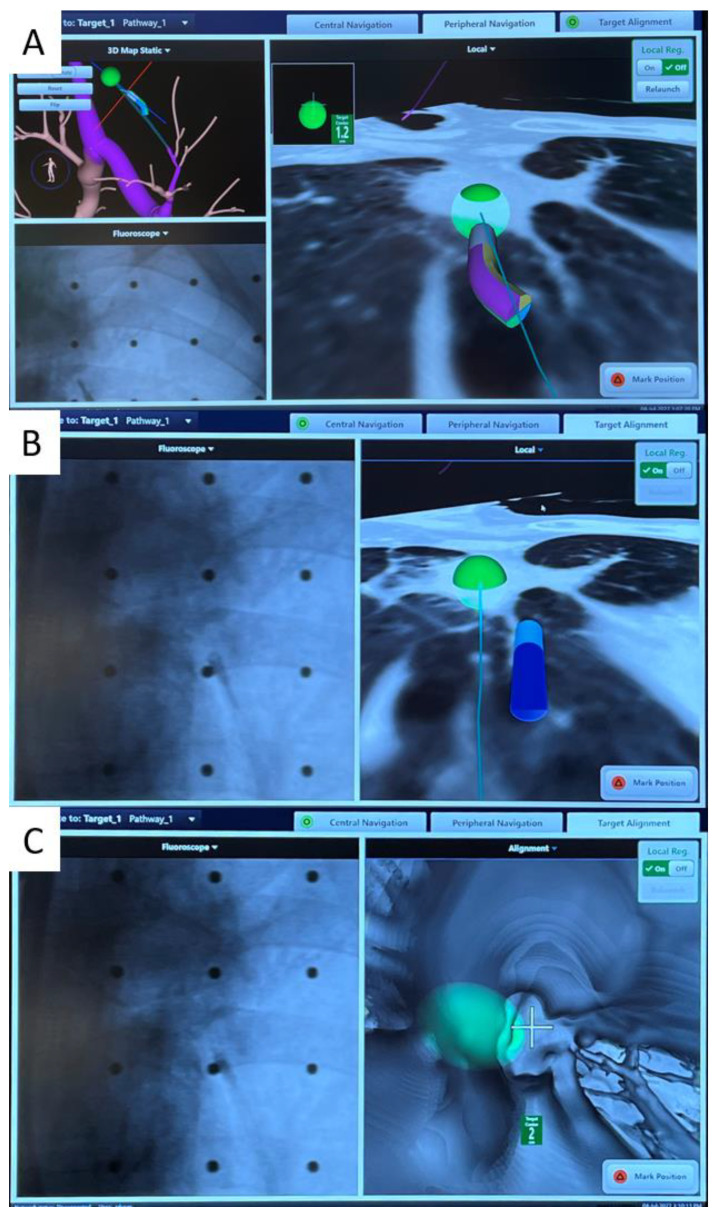
Images during ENB navigation using the Illumisite^TM^ platform. (**A**) Tip of the locatable guide pointing directly at the lesion (represented by the green ball) after peripheral navigation. In (**B**), the locatable guide was removed, but there were positional sensors embedded in the extended working channel which is shown pointing slightly off-center at the lesion. In (**C**), the crosshair is shown, pointing slightly off-center at the lesion which would require further manipulation to ensure target view, while the tip of the catheter is shown to be 2 cm from the target lesion.

**Figure 3 cancers-15-01068-f003:**
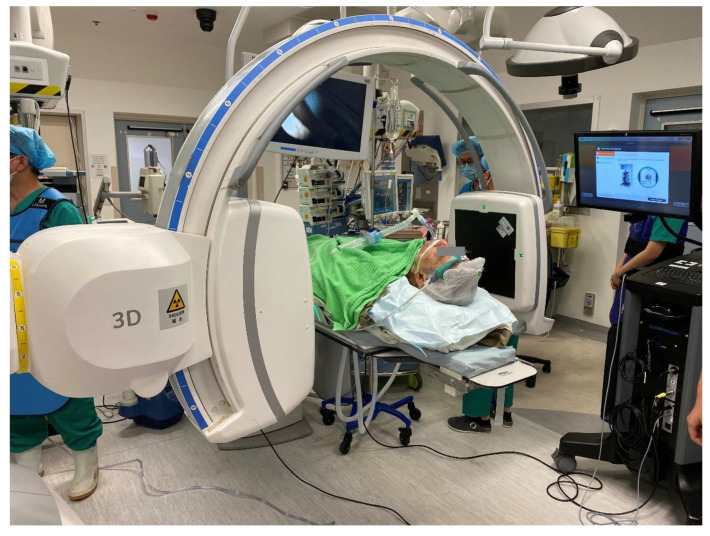
Picture of the operating room setup during Cios Spin^®^ mobile C-arm image acquisition.

**Figure 4 cancers-15-01068-f004:**
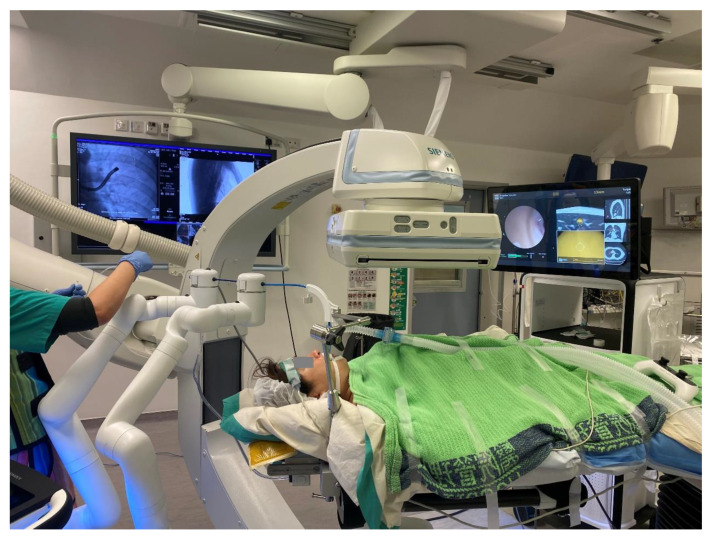
An example of robotic-assisted bronchoscopy by Auris Monarch^TM^ with cone beam CT in a hybrid operating room. This improves the accuracy of navigation, biopsy, or local treatment.

**Figure 5 cancers-15-01068-f005:**
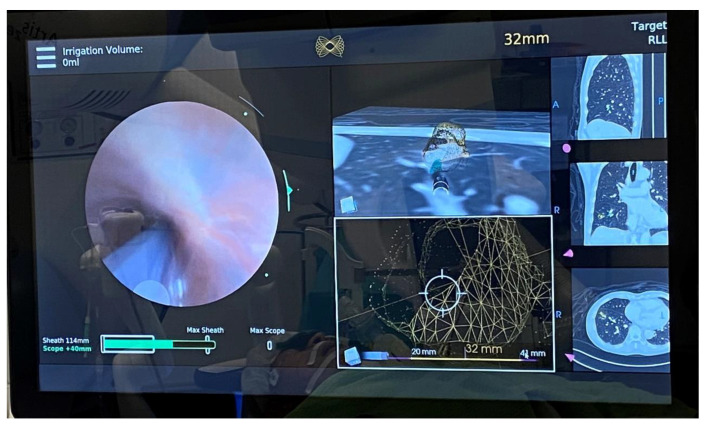
During robotic-assisted bronchoscopy by Auris Monarch^TM^, multiple panels can be visualized simultaneously: the left-hand panel shows the real-time bronchoscopic view, the middle ones show the bull’s eye view with the centered crosshair and the lesion (in yellow); the right-hand panels show the preoperative CT scan images in three perpendicular axes.

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
