# Peer review of "Transbronchial Techniques for Lung Cancer Treatment: Where Are We Now?"

_cancers, 2023, doi:10.3390/cancers15041068_

Round 1
Reviewer 1 Report
I would thank the authors for this interesting review cornering the innovative transbronchial treatment for lung nodules. However, there are some issues that I think should be addressed.
- the authors identify the transbronchial ablative treatment as a viable alternative to surgical resection, SBRT and percutaneous ablation. Nevertheless, surgical resection allows obtaining a post-operative pathological examination with molecular characterization, useful in case of the following progression of the disease. Do the authors suggest the achievement of pre-procedure diagnosis or during the ablative procedure? Missed diagnosis may be considered a limit of ablative treatment, this should be discussed in the paper.
- the description of the authors’ experience with transbronchial MWA should be resumed (line 125-137), avoiding reporting too many details, as was done for the other reported studies.
- the description of the different ablative systems is exhaustive and well-detailed. At the end, a summary of the characteristics of the different techniques, reporting their eventual pros and cons, should be useful
Reviewer 2 Report
Dear Doctor Chan, dear co-authors,
I truly enjoyed reading your manuscript. I learnt a lot from it.
I have a number of edits/suggestions, which I hope you will find of good use.
You will find line numbers when appropriate, and the use of italic type to help you identify where I stepped-in.
Content:
1) Title: I would write: Transbronchial Techniques for Lung Cancer Treatment: Where Are We Now? – You do not need to use the word ‘local’ in your title, as the transbronchial technique is indeed a local approach by definition. Also, by using Lung Cancer Treatment you can avoid using ‘Therapy’ (the validity of which, moreover, is to be verified with more data).
2) Simple summary: try to add a key aspect of your review. Something like: […] transbronchial therapies are being developed. This gives substantial hope to high-risk patients, as they would be otherwise excluded from currently established forms of local treatment.
3) Keywords: here is the place where I would definitely use ‘Local Treatment’. I would also delete ‘lung ablation’, and use only ‘Transbronchial ablation’ editing it as ‘Transbronchial Ablation Techniques’. I would also add ‘Lung Cancer Screening’.
4) 31: ‘Limited’ shall become sublobar. I would also delete ‘formal’.
5) 32-35: I would add references, or percentages, to specify ‘more and more small lung nodules’ and ‘many of which’.
6) 34: I would say: […] potentially eradicable with other, less invasive, forms of local treatment rather than surgery.
7) 40-41: I would say […] demonstrated a decreased mortality by 26% in high-risk men and up to 61% in high-risk women over a 10-year period by comparing low dose CT screening to no screening. à The NELSON trial, indeed, provided the basis for gender differences, so I believe you have a chance here to, at least, mention it. You can keep my sentence in your paper, if you like it.
8) 43: I would say […] which may become the best candidates
9) 46-47: I would say: Very often in Hong Kong
10) 139: I think it is important to briefly mention, right here in the text, the other types of instruments/devices that were used by your Chinese colleagues
11) 163-165: here is the first time in the manuscript that you mention tissue sampling. Quick question: when you do MWA with ENB at your institution, do you deal with tissue sampling at all? If so, when and how do you attempt cells/tissue collection? Do you use EBUS-TBNA +/- radial probe with ROSE for diagnosis and then, within the same session, you proceed with the ablation? Or do you do it in two different sessions? Or, do you simply skip the diagnosis because you are clinically confident enough that the nodule is malignant? Given the fact that SBRT and ablation are techniques that can indeed suffer from the lack of cells/tissue for diagnosis (especially for those who may need subsequent systemic treatments), I would try to clearly state somewhere in your paper (ideally early on) how you believe that local treatments shall relate to p-diagnosis (and viceversa).
12) 180: ‘previous platforms’ & 250-251: ‘previous non-robo tic systems’: can you elaborate a bit on these? I think it is important to provide the reader with a quick overview of what’s out there, given the fact that it is/has been used on patients.
13) 224: The title itself is not a form of local treatment (which is the focus of your review). I would modify it like this: Robotic bronchoscopy to improve transbronchial ablation accuracy & efficacy
14) 283-284: How did you assess the significant remaining viable tumour cells?
Style:
1) Numerical single digits (below 10) that refer to nouns shall be written in letters (3 branches = three branches; forced expiratory volume in 1 second = in the first second or in one second; at 4 years = at four years; see also lines 132, 133, 137, 152, 195, 281-283). This suggestion does not apply to ‘phase 3’ or single digits with metric systems (like ‘2cm’) or that are bound to words (like ‘5-year survival’). With regard to ‘stage 1’ (line 78) I would write ‘stage I’ instead, just like you correctly did in line 85.
2) Punctuation: semicolons in your abstract shall be changed with commas (as you did in your sterile copy-pasted version of the Conclusions).
3) Figure 1: very poor image quality, especially on the print-out. If you could increase it, it would be great (figures are a powerful tool to deliver clear messages)
4) Figure 1 and line 151: here are the first times that ‘CBCT’ appears. Spell it out in the text à Cone-beam computed tomography systems (CBCT)
5) Figure 2C: is evidently misaligned: adjust it, for the sake of good pic presentation
6) Figure 4: I would re-write it like this: An example of robotic bronchoscopy by Auris Monarch used with cone beam CT in a hybrid operating room. This improves the accuracy of navigation, biopsy and local treatment.
7) It strikes and disappoints me to see that the conclusion paragraph is just a sterile copy-paste of the introduction. Please, just do something better. Convey your vision.
English:
1) 16: I would write: In this review we discuss the state-of-the art of transbronchial techniques for lung cancer treatment.
2) 31: rate à rates
3) 45: Try to be consistent with your wording throughout your paper. You use ‘carcinoma’ here. I would simply keep on using ‘cancer’, instead.
4) 48: […] with a predisposition
5) 54-55: […] making them unsuitable for major pulmonary resection hence better candidates for local treatment.
6) 63: […] offering an opportunity
7) 71: Try to avoid repetitions throughout your paper. So, delete ‘small sized’ (you are already saying, a few words later, ‘less than 2cm’)
8) 74: […] than in the lobectomy group
9) 87:
a. ‘and disease-free’ (delete ‘and’, in favor of ‘a’) à a disease-free survival of 26%
b. […] and an overall survival
10) 88-89: […] still carries risks such as intra- and post-operative surgical complications and radiation pneumonitis, respectively.
11) 90: […] following the success
12) 97-98:
a. […] of the transbronchial
b. ‘and hence’: delete ‘and’
13) 102: […] certain parts of the lung
14) 103-104:
a. […] difficult or even dangerous with the percutaneous access
b. […] near the mediastinal pleura
c. […] shielded by the scapula
15) 105: I would delete ‘Taking the above into consideration’ and simply start the sentence by saying: Our institute is one of the first to […]
16) 106-108: I would say ‘We prefer microwave energy because, unlike thermal energy, it directly heats tissues to lethal temperatures (>150 °C) through dielectric hysteresis(24), therefore it is independent from […]’
17) 111: Patients treated with this novel technique at our center had […]
18) 149: […] far from being perfect
19) 162: […] in the tip of the extended
20) 168: […] represented by the green ball
21) 182: […] of the tip of the ablation catheter
22) Figure 3: Picture of the operating room […]
23) 212: […] into account a 5mm margin
24) 222: […] be injected into tumour
25) 234-244: I would re-phrase most of the paragraph like this: ‘The inner bronchoscope has superior maneuverability than a conventional bronchoscope due to a 4-way steering control. While the inner scope is being advanced, structural support is provided by the RAB system hence reducing the likelihood of scope prolapse into proximal airways. The Ion RAB comprises of an articulated catheter with a 3.5mm outer diameter and a 1.8mm removable visual probe […] on the location of the catheter tip. Shape sensing technology allows positional adjustments compensating for a degree of deformation created while relatively rigid instruments pass through the bronchoscope’.
26) 248: Delete ‘Excellent’ (I think it does not add anything to your sentence). I think we can all appreciate that a navigation success up to 97% is an excellent result.
27) 250-251: I would re-phrase like this: […] the diagnostic yield was higher than that reported using previous non-robotic systems (54 vs. 31-44%, respectively)
28) 253-257: I would re-phrase like this: […] reported a 97.5% navigational success and a 91% nodule localization by radial EBUS, […]. RAB provides the ability to control the distal end of the scope through multiple active articulation points of the scope, thus allowing enhancing instruments maneuverability.
29) 259:
a. […] room it further
b. […] accuracy of ablation
30) Figure 5: I would re-phrase like this: During the robotic bronchoscopy by Auris Monarch, multiple panels can be visualized simultaneously: the left hand one shows real-time the bronchoscopic view; the middle ones show the bull’s eye view with the centered crosshair and the lesion (in yellow); the right hand panels show the pre-operative CT scan images in three perpendicular axes.
31) 269: Temperature can reach up to -40°C […]
32) 270: I would re-phrase like this: Cryoablation carries multiple advantages: the sized of the ablation zone are generally bigger and independent from impedance, bronchovascular structures are better preserved, no tissue charring is created and excellent visibility of the iceball on CT is obtainable(57-59).
33) 273-275:
a. […] approximately 25 vs 10 minutes for cryoablation and MWA, respectively.
b. Change ‘models’ to its singular form.
34) 284-285: BTVA may be inadequate as the sole cancer treatment modality, but it can be used […] ablated transbronchially with MWA, while the margins, satellite nodules and spread-through-air-space surrounding the nodule can be captured by TVA afterwards, thus […].
35) 291: […] around a catheter
36) 292: […] voltage crosses a threshold
37) 293: […] related to the formation
38) 298-299: I would add a full stop after […] vein stenosis), to make the sentence easier to read. Then, I would start the following sentence like this: Also, energy can be delivered […]
39) 306: I would add a full stop after […] swine models, and start the following sentence like this: Histologically, short-term […]
40) 309-310: […] without pleural disruption
41) 312: you shall add San Carlos, CA - USA
Author Response
Please see the attachement.

Round 2
Reviewer 1 Report
Dear authors,
I would like to congratulate you on the job you have done.
I have some suggestions:
In line 124, it would be better to write “The microwave energy, unlike radiofrequency, heats tissue directly to lethal temperatures > 150°C through dielectric hysteresis (24)…”.
It is advisable to avoid expressing personal opinions (“we prefer…”) and to limit self-citations.
Author Response
In line 124, it would be better to write “The microwave energy, unlike radiofrequency, heats tissue directly to lethal temperatures > 150°C through dielectric hysteresis (24)…”.
It is advisable to avoid expressing personal opinions (“we prefer…”) and to limit self-citations.
Reply and change to text: Thank you very much for the suggestion, and we have amended the manuscript as per your recommendation.